# Reduction of variability for the assessment of side effects of toxicants on honeybees and understanding drivers for colony development

**Magnus Wang***[�], **Thiemo Braasch**[�], **Christian Dietrich**[�]

WSC Scientific GmbH, Heidelberg, Germany

[�] These authors contributed equally to this work.
* magnus.wang@wsc-regexperts.com

**Data Availability Statement:** All relevant data are within the paper and its Supporting Information files.

## Abstract

The statistical power of studies for the assessment of side effects of toxicants on honeybees conducted according to current guidelines is often limited. A new test design and modified field methods have therefore been developed to decrease uncertainty and variability and to be able to detect small effects. The new test design comprises a monitoring phase (before the tunnel phase) for the selection of honeybee colonies and modified methods, which include assessments of colony strength, an evaluation of the cell content of all cells of hives using photos and digital analysis, and the use of video recordings for the assessment of foraging activity and forager mortality. With the proposed new study design and the modified field methods variability between hives was considerably reduced, which resulted in a marked reduction of the minimum detectable difference (MDD). This makes it possible to address the Specific Protection Goals defined by the European Food Safety Authority and to gain unprecedented insight into the development of hives and driving factors.

## Introduction

Currently, honeybee field and semi-field trials conducted to assess side effects by toxicants, such as pesticides, are based on visual assessments of colony strength and evaluations of brood success for a relatively small number of brood cells. The test guidance documents for the conduct of such studies [1, 2] propose that colony strength is estimated visually by estimation of the comb area covered by bees and, regarding brood development, the evaluation of at least 100 cells containing eggs is required. EFSA [3] proposes to consider at least 200 eggs. Based on a typical brood nest size of about 3000 to 6000 cells [4], this is still a rather small proportion. The assessment of colony strength based on visual estimation (e.g. using the Liebefelder methods [5]) seems to provide a rather useful average accuracy. However, it has not been evaluated which statistical test-power can actually be reached by visual estimations. Notably, test-power depends on the variability of a measurement rather than on the average. Other endpoints measured in honeybee trials are flight activity and mortality (e.g. from counts in dead bee traps). Also the measurement of these parameters is typically very approximate only and they often

**Funding:** This research was funded entirely by the commercial company WSC Scientific GmbH. The funder provided support in the form of salaries for authors [MW, TB, CD] and materials, but did not have any additional role in the study design, data collection and analysis, decision to publish, or preparation of the manuscript. The specific roles of these authors are articulated in the 'author contributions' section

**Competing interests:** [MW, TB, CD] are employed at WSC Scientific GmbH, which, amongst others, develops scientific software including software for the evaluation of honeybee trials. The purpose of this study was, however, not to promote such software, which contributes only marginally to the turnover of the company, but to develop better study design for the risk assessment. This does not alter our adherence to PLOS ONE policies on sharing data and materials.

represent a small snapshot of what happens over a day. For example, flight activity is measured at three locations in 1 m squares for a few seconds [1]. Also the area covered by dead bee traps or linen sheets for counting dead bees is limited. Considering that honeybees may fly throughout the day and that in a semi-field study a tunnel should measure at least 40 m$^2$, these measurements provide only a very small snapshot of the actual flight activity and the estimation of the proportion of foragers is not possible (since it is unknown how many bees foraged and how many bees died outside of traps or linen sheets). The standard methods used in OECD (2007) [1] and OEPP/EPPO (2010) [2] clearly have their justification and are very useful to routinely evaluate potential effects of chemical substances. But to reach a better understanding of the robustness of an evaluation and of why colonies develop the way they do, it would be helpful to obtain more data and more accurate data. For example, it would be helpful to understand which stores and brood are exactly available in a hive and how these affect the colony's development or which proportion of foragers relative to all foragers dies after the application of a test substance. Also when a higher statistical power is desired new field methodology is needed. In fact, one key issue of recent evaluations of pesticides by authorities was the limited test-power that makes it impossible to detect small effects [6]. Recently, EFSA (2013) [3] proposed that field or semi-field studies should be able to detect 7% effect size regarding colony strength (a value of 7% has previously been proposed to represent a negligible effect size regarding forager mortality [7] and has then been adopted as a value representing a negligible effect level for colony strength; [3]).

The reasons why a high test-power is hard to reach in practice include the uncertainty of current methods (e.g. visual estimation) and analysis of a limited fraction of a hive (e.g. number of cells). But also the high variability introduced by current test designs (e.g. selection of hives) is a reason for an insufficient test-power. To overcome the high variability among honey bee colonies, Delaplane et al. [8] described methods to obtain equalized honeybee colonies. This included the so-called "classical objective mode", which is a synthesis of methods presented by Harbo [9, 10, 11, 12] and Delaplane and Harbo [13]. In this method empty hives are pre-stocked with brood, empty combs, syrup feeders and a caged queen. Then worker bees are added. This mode was adapted for investigations about *Varroa destructor* e.g. in [14, 15, 16, 17], and about colony growth [18]. The second mode was called the "Shook swarm objective mode" by Delaplane et al. [8]. In this method workers of the same origin are put into new and empty hives without brood. Later the origin queens are added.

Also in the present study the aim was to reduce variability. However, this was not done by manipulation of colonies, but by (i) a selection of colonies after a monitoring phase (before the tunnel phase) approximately lasting four weeks, (ii) the assessment of entire hives (including all cells of a hive, instead of tracking the development in a selected number of brood cells) and (iii) by applying different methods for measuring colony strength and mortality, in order to see which is the most sensitive method. The key principle is the reduction of variability by selection of hives from a larger subset of hives (see i). In the following we refer to this study design as 'low uncertainty and variability test' (LUV test). This methodology was tested in a semi-field study conducted under Good Laboratory Practice (GLP). Results from our semi-field study demonstrate that variability of both field and semi-field trials can be decreased significantly with the proposed new test design and methodology.

## Material and methods

### Test design

The study design was a modification of OECD (2007) [1]. The study was conducted under Good Laboratory Practice (GLP) in a large oilseed rape field near Heidelberg, Germany. Since

the aim of the semi-field study was not to assess the toxicity of a test substance but to test new field methodology, only a control group (applying tap water) and a reference group were used (applying 400 g/ha dimethoate). The study was divided into three phases: 1. An about four week monitoring phase before the tunnel phase (during this time colonies were kept at the bee keeping facility), during which colonies were assessed regarding colony size, brood development and food stores and mortality in dead bee traps. 2. A ten-day tunnel phase, during which colonies were placed in tunnels ($100 \text{ m}^2$) on an oilseed rape field in full bloom. Application was conducted two days after bees were in tunnels. 3. An about four week post-tunnel phase, during which monitoring continued (during this time colonies were again kept at the bee keeping facility). Drone brood was removed when capped. Throughout the whole study the following endpoints were measured (see below for details on the methods): colony strength, content of all cells of all hives (incl. brood, nectar, pollen), dead bees and larvae in traps. Foraging activity by visual assessment was only conducted in the tunnel phase and forager activity and mortality measurements by videography were only conducted in the tunnel and post-tunnel phase.

**Step 1: Four week monitoring phase.** The study was started with 16 colonies with sister queens (Carniolan honeybees) obtained from a commercial beekeeper. During the monitoring phase (before the tunnel phase) five colonies were excluded because two colonies did not contain related sister queens, one hive had no queen at all, one colony had slightly elevated *Varroa destructor* infestation and bees of one hive were very aggressive. The infestation of the hives with *Varroa destructor* was assessed via counting natural mite drops. All hives were compared regarding their *Varroa* counts. Then the hive with the highest *Varroa* infection was eliminated as this infection rate was above the acceptable *Varroa* infection level. Hence, at the end of the monitoring phase eleven colonies were considered for the selection for the tunnel phase.

**Selection of colonies at the end of the monitoring phase.** From these eleven colonies eight colonies were selected and randomly assigned to the control and the reference group (four colonies per group). The selection had the aim of achieving colonies of similar strength during the tunnel phase. This same rationale is used in many laboratory toxicity trials, where very 'similar' animals, e.g. of similar age and strain are selected in order to decrease variability (and increase test-power). This was achieved based on the development of colony size throughout the pre-tunnel monitoring phase (as measured by weight), the number of capped brood cells per hive and similar mortality (measured in dead bee traps).

**Step 2: A ten-day tunnel phase.** After selection of colonies, colonies were placed in tunnels. Colonies remained in tunnels for ten days. Toxicant application (reference substance) was conducted two days after the start of the tunnel phase. This step (tunnel phase) was identical to the procedure described in OECD text 75.

**Step 3: Post-tunnel phase.** After the tunnel phase colonies were relocated to their original location and monitored for about four additional weeks. This phase allowed to assess potential recovery of colonies.

## Estimation of colony size

The limited accuracy of visual estimation of colony strength following the Liebefelder method results in some uncertainty which decreases test-power [19, 20]. Since this variability alone can be a reason for not reaching a high test-power, colony size was measured based on two alternative methods in addition to visual estimation: 1. Weighing of hives with and without bees. 2. Photography of bees on frames. Colony size was estimated with all three methods during all phases of the study.

Weighing: Hives were closed in the evening after flight activity. In the next morning closed hives were weighed and afterwards entrances were opened. Subsequently, all parts of the hives

were weighed without bees by gently brushing bees off. A weight of 100 mg per adult bee was considered to calculate the number of bees. The number of all bees represents all adult bees including all foragers.

Adult bee photography: All frames of all hives were photographed to count the number of adult bees. Photography was conducted simultaneously for all hives to exclude any bias due to changes of weather, which might affect foraging activity. The number of bees was counted automatically in photos using the software HoneybeeComplete 6.0 (WSC Scientific GmbH). This number represents all adult bees without active foragers. The accuracy of automated counting had previously been validated (accuracy was ~80–90%, automatically counted vs. real no. of bees). Correction factors reflecting this accuracy were applied.

Finally, for comparison also visual estimations of colony size were conducted following the Liebefelder method [5].

## Estimation of brood success and food stores

Since the evaluation of a limited number of brood cells as proposed in OCED (2007) [1] or EFSA (2013) [3] results in an uncertainty that increases the measured variability between brood termination rates between hives [21], all frames of the hives were photographed with a 36 MP camera and the content of all cells (usually more than 3000 brood cells, more than 100 000 cells per hive) was evaluated. With these photos the development of brood and the amount of nectar and pollen stores was assessed. Cells and cell content were recognized with the software HoneybeeComplete 6.0 and the content of cells was manually verified and corrected when necessary. Also the brood development was evaluated with this software. Brood photography was conducted following the time intervals proposed in OECD (2007) [1]. In addition, brood photography was conducted in weekly intervals during monitoring and after the end of photography according to OECD (2007) [1].

## Flight activity and forager mortality

Measurements of flight activity were planned in weekly intervals during and after the tunnel phase using video recordings of the entrances of all hives over the entire activity phase (from dawn to dusk; see S1 File). Actual dates varied by 1–2 days depending on weather conditions. Recordings were processed with the software VideoCounter 1.1 (WSC Scientific GmbH) to count the number of bees exiting and entering the hives. The software was previously validated regarding the accuracy (which was 108.7% for bees entering the hive and 82.7% for bees leaving the hive) of counting and correction factors were applied to obtain corrected counts. These counts reflect foraging activity and from these counts forager mortality can be estimated by subtracting the daily number of entering bees from the number of leaving bees. In addition, flight activity was also assessed in three 1 m squares per colony as proposed in OECD (2007) [1], but longer observation periods of 30 seconds were used.

## Dead bee counting according to OECD 75

Dead bees were counted using dead bee traps (type underbasket) during all phases of the study. During the tunnel phase dead bees were additionally counted on 80 cm wide sheets placed in the centre, front and end of tunnels.

## Weather data

Weather data, including temperature and precipitation, were obtained from the weather station nearest to the study field (3.1 km distance to the study field; non-GLP).

## Statistical analysis

**Limit of detection.** To evaluate which effect size could be detected the minimum detectable difference (MDD) was calculated according to Brock et al. [22]. MDD is based on the assumption of normal data distribution, which may not be expected with regard to colony size. However, MDD was still used as it is a measure that is already established in the risk assessment of pesticides (e.g. [23]) and that is frequently requested by authorities. MDD calculations were conducted for the number of adults obtained from weight measurements, from photography and by visual estimations (Liebefelder method).

**Hypothesis testing.** Differences of colony strength between the control and the reference were assessed statistically using a t-test (as there were only two test groups; significance level $\alpha$ = 0.05). Before conducting this test, data were checked for normality using Shapiro-Wilks Test. Homogeneity of variances was tested using Bartlett's test. A significance level of $\alpha$ = 0.05 was considered. The statistical analysis was conducted in R [24].

**Analysis of factors determining brood termination.** Brood termination can be described as a function of constitutive in-hive variables such as the number of pollen cells. To evaluate the relevance of these variables, generalized linear models (GLMs) were generated in R [24, 25] and compared on the basis of AICc (AIC correction for small sample sizes). Since the dependent variable $BTR_{egg}$ is a proportion, a beta distribution was assumed using logit link function. $BTR_{egg}$ of control group (four hives) measured at nine points in time was considered. Besides the single variables also their quadratic terms and two-way interactions were considered as linear components in the models (e.g. $y = a_1x_1 + a_2x_2 + a_3x_3 + a_4x_4 + a_5$ with $x_3 = z_1^2$ and $x_4 = z_2z_3$). Furthermore, highly correlated variables ($r > 0.7$) were excluded in advance to avoid multiple incorporation of the same effect and to enable a proper regression analysis.

# Results

## Pre-monitoring

The pre-monitoring of the study started with 16 colonies obtained from a commercial beekeeper and labelled as colonies with sister queens from 2016. Over a period of 32 days they were assessed approximately weekly (depending on weather conditions) to finally select eight very similar colonies (e.g. regarding colony size, brood, *Varroa destructor* counts). During this pre-monitoring period two colonies were excluded as the queens were evidently not sister queens (wrong colour marks). Furthermore, one was excluded due to slightly elevated *Varroa destructor* infestation, one contained very aggressive bees and in one colony no queen was found. After exclusion of these colonies eleven colonies remained for selection of colonies for the tunnel phase. Of these eight were selected for the tunnel phase, which were expected to be most equal during the tunnel phase with respect to colony strength based on the number of adult bees and capped brood cells and mortality. By the selection of four hives per test group after the pre-monitoring period MDD and mean CV (coefficient of variation) values of the control and reference were reduced from 29.4% (MDD) and 19.4% (CV) to 10.8% and 7.2% for estimations by weight (Fig 1, Table 1), from 19.5% (MDD) and 12.4% (CV) to 13.7% and 8.9% for estimations by adult photography and from 20.3% (MDD) and 12.8% (CV) to 13.7% and 9.0% for visual estimations. This demonstrates that colony size estimations based on colony weight or photography would be able to detect small effects on population sizes of colonies of almost 10%, even though only four replicates (tunnels) were used per test group. For detailed information of means and standard deviations underlying the power analysis see S1-S3 Tables in S1 File. We also compared MDD of colony strength by weight on the day of selection of colonies under the assumption of randomly sampling four colonies for each test

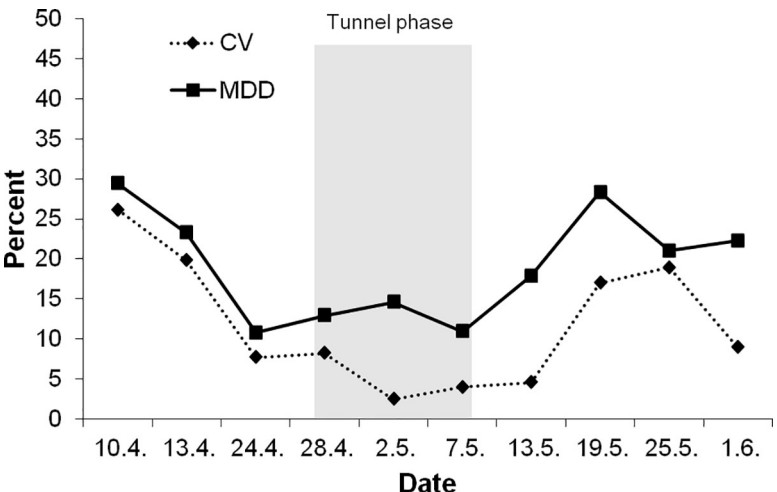

**Fig 1. Minimum detectable difference (MDD) [%] and coefficient of variation (CV) [%] for the number of adults obtained from weight measurements.**

group by Monte Carlo analysis (with replacement). When randomly choosing hives MDD was on average 21.6%. I.e. the selection procedure achieved a reduction of MDD by a factor of about two (10.8% by the selection procedure vs. 21.6% by randomly choosing hives).

## Tunnel phase

After the selection of the colonies, four hives were randomly assigned to each the control group and the group in which a toxicant was applied (reference group), respectively and they were placed into the tunnels. After application (at the beginning of the tunnel phase), colony size in the reference dropped by about 21% (Fig 2). This decrease of the population size of the reference hives was clearly detectable by weight measurements and adult photography (Table 1). When evaluating colony size from visual estimations the effect was statistically not significant. Notably, MDD for colony size obtained from weighting and photography remained low throughout the tunnel phase.

**Table 1. Minimum detectable difference and p-values of t-tests of the effect of toxicant exposure on colony strength estimated by weight, adult bee photography and visual estimation between the control and the reference.**

| | Pre-tunnel phase | | | Tunnel phase | | | Post-tunnel phase | | | |
|---|---|---|---|---|---|---|---|---|---|---|
| **Date** | **10.04** | **13.04** | **24.04** | **28.04** | **02.05** | **07.05** | **13.05** | **19.05** | **25.05.** | **01.06.** |
| **Minimum detectable difference (MDD, %)** | | | | | | | | | | |
| **Weight** | 29.4 | 23.2 | 10.8 | 12.9 | 14.6 | 10.9 | 17.9 | 28.3 | 21.0 | 22.3 |
| **Photo-graphy** | 19.5 | 25.2 | 13.7 | 14.7 | 14.3 | 18.1 | 20.7 | 18.6 | 18.4 | 20.0 |
| **Visual estima-tion** | 20.3 | 25.2 | 13.7 | 18.7 | 18.3 | 24.6 | 19.5 | 17.7 | 27.1 | 19.7 |
| **p-values from t-tests** | | | | | | | | | | |
| **Weight** | n/a | n/a | n/a | 1.000 | 0.032* | 0.019* | 0.171 | 0.458 | 0.029* | 0.724 |
| **Photo-graphy** | n/a | n/a | n/a | 0.997 | 0.016* | 0.100 | 0.397 | 0.376 | 0.130 | 0.128 |
| **Visual estima-tion** | n/a | n/a | n/a | 0.998 | 0.179 | 0.217 | 0.472 | 0.214 | 0.196 | 0.407 |

* Significant difference between control and reference hives at $p < 0.05$.

Data did not deviate significantly from normality and variances were homogeneous. n/a = not applicable.

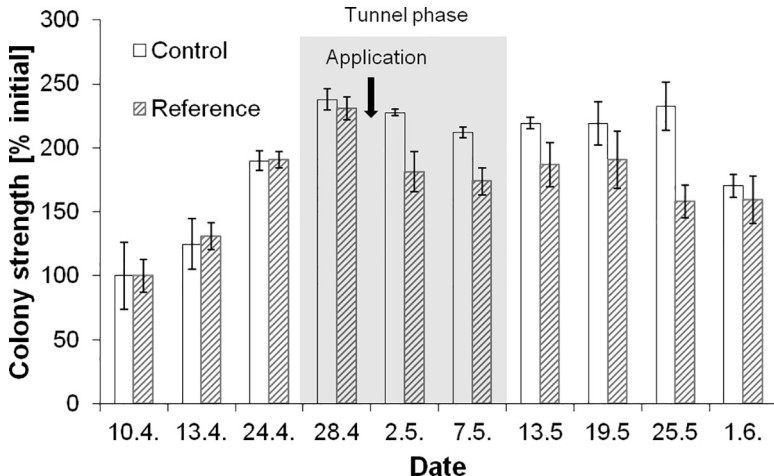

**Fig 2. Colony strength [% initial] obtained from the estimations of the numbers of adult bees measured by weight.** Error bars reflect the standard deviation between colonies (data had been normalised to the mean of the control group).

Application of the reference substance also resulted in increased forager mortality (Fig 3) estimated from automated video counts and in a high number of dead bees in dead bee traps and sheets (S2 Fig, S4-S6 Tables in S1 File). Flight activity obtained from automated counts of bees leaving the hive was reduced by 34% after the application during the tunnel phase, whereas the flight activity obtained by observations of 1 m squares in the crop was reduced by 91% after the application during the tunnel phase. This difference could be due to bees deciding not to forage in the sprayed crop after an initial assessment of the environment.

## Post-tunnel phase

During the post-tunnel phase both control and reference colonies showed a similar weather related development. One month after application effects on colony size had disappeared (Fig 2, Table 1).

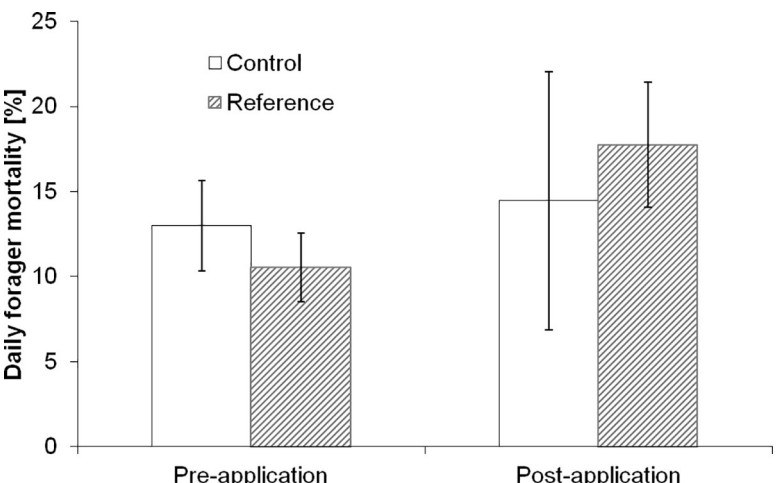

**Fig 3. Daily forager mortality in the tunnel before and after the application.** Error bars indicate one standard deviation.

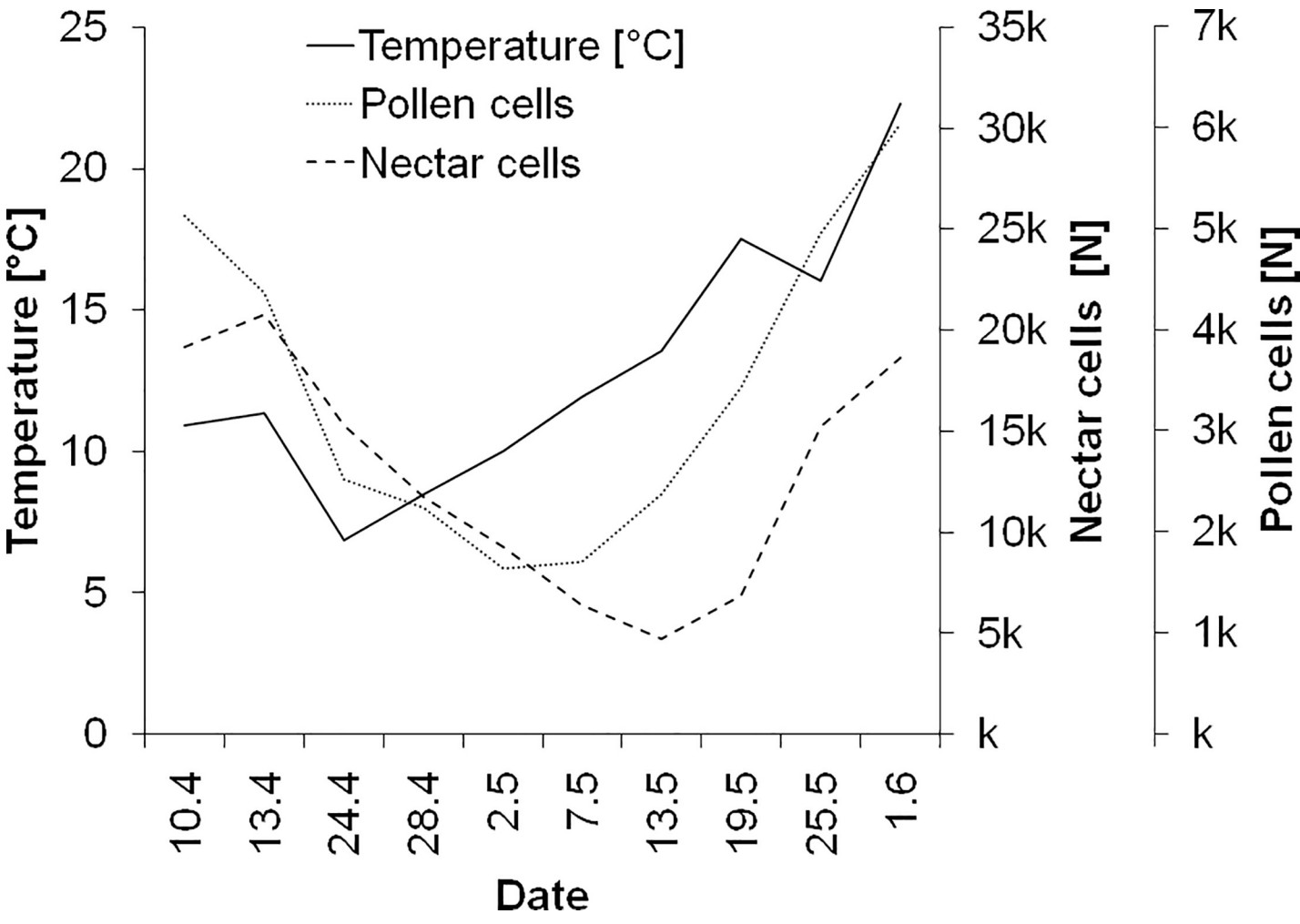

**Fig 4. Relation of the amount of pollen cells of all control hives to temperature during the study.**

### Brood development, food stores and influence of weather

Photos of all frames of all hives were taken during all phases of the study. Therefore, brood development could be evaluated continuously over the study period. The brood development of both control and reference hives (amount of eggs, old larvae, young larvae und pupae) showed a similar trend during the whole study. Weather had a pronounced effect of the brood development in both control and reference hives, which was more pronounced than the treatment with the reference substance. Low temperatures (<12˚C) coincided with marked reduction of nectar and pollen stores (Fig 4, shown for control hives).

Brood termination mainly occurred during the egg stage, i.e. very early during the development (up to 83% of the total termination occurred at the egg stage). This early termination of eggs ($BTR_{egg}$) was highest when the ratio of pollen cells to open brood cells dropped below 1.4 (Fig 5 left).

To identify parameters which mostly affect $BTR_{egg}$ more systematically, generalized linear models (GLMs) were used. A complete list of parameters is provided in the supplementary material. Prior to the model analysis the data was checked for pseudo-replicates due to repeated measurements in the same hives. However, the termination rate refers to the eggs rather than to the hives and the entity of eggs completely changed from one measurement to

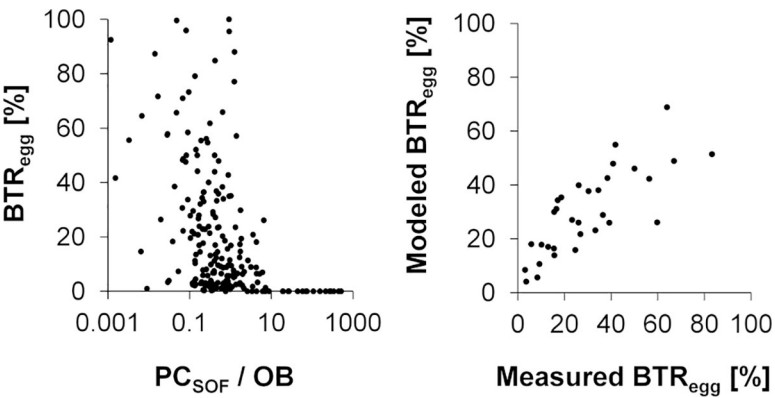

**Fig 5. Left: Dependence of BTR$_{egg}$ on the ratio of pollen cells on same or opposite frame to open brood cells.** Right: Model fit of a best fit GLM for BTR$_{egg}$ considering impact of capped and open brood cells and pollen cells (model no. 3 in Table 2).

another. Furthermore, there was no significant correlation between the BTR$_{egg}$ and the hive index or the time of measurement (S3 Fig in S1 File). Hence, the BTR$_{egg}$ measurements can be considered as independent.

The visual analysis of the hive frames indicated that brood termination depended on the spatial distance between pollen cells and brood cells (Fig 6). Statistical analysis confirmed the visual analysis and identified the number of pollen cells on the same or opposite frame and the distance from the hive center as the most important drivers of the BTR$_{egg}$ inside the hive (S3 Fig in S1 File).

When evaluating the hive content without taking account of this spatial information (i.e. ignoring whether e.g. pollen is available near brood cells or far away), the best models describing BTR$_{egg}$ included the parameters capped brood, open brood and pollen cells. When analyzing the hive content taking spatial information into account, a high BTR$_{egg}$ coincided with a low number of pollen cells on the same or opposite frame, a high number of capped brood on the same or opposite frame (possibly indicating a high recent consumption of pollen of previously open brood that is now capped) and a low ratio of pollen cells per open brood cells and a low number of open brood (S3 Fig in S1 File). Hence, the spatial distribution of pollen seems to play an important role for brood success.

## Discussion and conclusions

Currently used honeybee field and semi-field trials conducted to assess side effects by toxicants have been criticised of having a relatively low statistical power, resulting from the inherent

**Table 2. Generalized linear models (GLMs) describing the impact of factors on brood termination rate BTR$_{egg}$.**

| GLM for identification of BTR$_{egg}$ drivers | | | | |
|---|---|---|---|---|
| Model | N param. | Explanatory variables | AICc | ΔAICc |
| 1 | 2 | CB + OB | -36.30 | +5.28 |
| 2 | 3 | CB + OB + OB$^2$ | -38.84 | +2.74 |
| 3 | 4 | CB + OB + PC + CB*PC | -41.58 | 0 |
| 4 | 5 | Same as model 3 | -41.58 | 0 |

A list of evaluated parameters is available in the supplementary material. The explanatory variables are CB = Capped Brood, OB = Open Brood, PC = Pollen Cells. The ΔAICc references to the lowest AICc among all models. Since the best model with five parameters equals the best model with four parameters no more models with five parameters are listed. Exceeding the number of four model parameters will lead to an overparameterization.

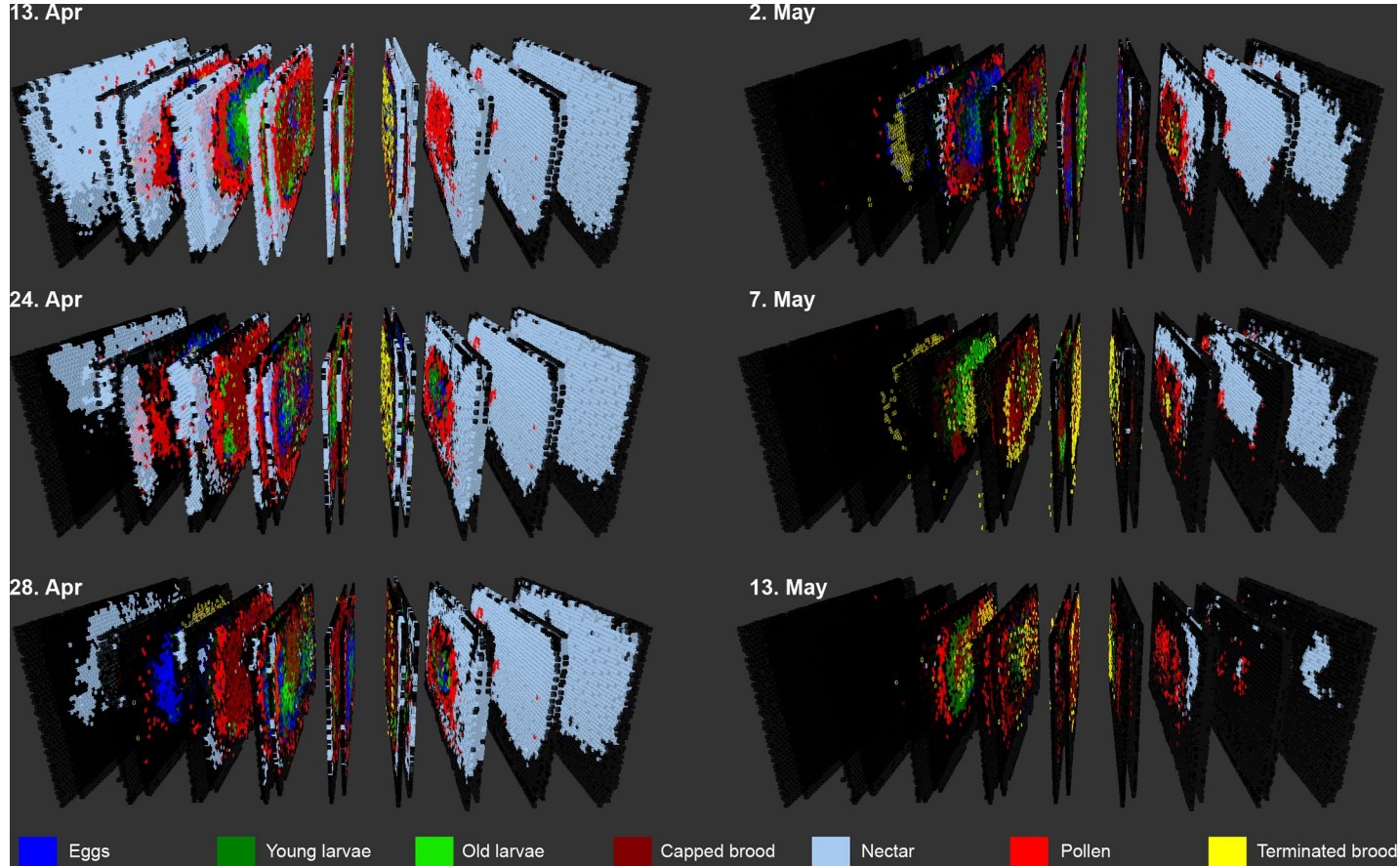

| Eggs | Young larvae | Old larvae | Capped brood | Nectar | Pollen | Terminated brood |

**Fig 6. 3D representation of the bottom body of a colony hive (hive 17–5).** During a period of cold weather nectar and pollen stores were gradually depleted. It can also be seen that brood termination (shown in yellow) was sometimes high, when pollen stores on the same or opposite side of a frame were low (see e.g. 13[th] and 24[th] April). Even though pollen was available elsewhere in the hive, it seems that this did not prevent termination.

variability of honeybee colonies but also from field methodology [3, 6]. In the past, this has been addressed by equalizing honeybee colonies before the start of a test. For example, Delaplane et al. [8] presented two different variations of a method to reduce variation among honey bee colonies regarding adult bees, brood, mites and food at the beginning of experiments. Both modes described by Delaplane et al. [8] involve significant manipulation of colonies (and possibly a mix of bees from different queens may be obtained when brood combs are combined). In the study design considered in our study, instead of equalizing hives we started with a larger number of unmanipulated hives and tracked their development over about a month in order to see how they perform over time. Then colonies were selected for the test, which were not only of similar size, but which developed similarly throughout the pre-monitoring phase (taking account of mortality and capped brood) and which were hence expected to perform similarly during the tunnel phase. We used sister queens, but since no new colonies were set up using combs from different hives, in our trial all bees can be considered to be offspring of their queen, which reduces genetic variation (for a comparison of the present study design with a conventional OECD 75 test, see also S10 Table in S1 File).

Apart from reducing variability by the selection of colonies after pre-monitoring, another aim of this study was to test different methods with regard to their variability and uncertainty, in order to identify those methods, which have the potential to decrease variability in field and

semi-field trials. This had been preceded by an assessment the sources of uncertainty and variability [19, 26]. This included the measurement of colony size by weight, adult bee photography and the assessment of brood development and food stores with in-hive photography. The determination of colony size by weight has previously been described by Delaplane et al. [8]. But in contrast to Delaplane et al. [8] weighing of the colonies was not only done at the end of the experiment but during the whole course of our study. Evaluation of bee brood via photography has been described by various authors (e.g. [26, 27, 28, 29, 30]). However, to our knowledge the present study is the first one using digital analysis of photos of all combs of all hives during the whole study period (including not only brood cells, but also any other type of cell), instead of selecting a subset of 100 or 200 brood cells only [1, 3]. The use of photography to count adults has previously also been used by Cutler et al. [31]. A new method applied in this study was the assessment of flight activity and forager mortality with the help of automated video counts instead of using 1 m observation squares, dead bee traps and linen sheets, which cover forager activity and forager mortality only partially.

The presented study design together with the tested measurement methods resulted in a considerable reduction of MDDs of colony strength (except for visual estimation, for which MDD were generally highest). In particular, the assessment of colony strength obtained from weight measurements made it possible to detect effects of about 10%, despite of the low number of only four replicates per test group (MDD values before the application and during the tunnel phase after the application). Therefore, by using this method it was possible to detect rather small differences between hives. If the number of control and reference hives were doubled for estimations by weight (i.e. eight control hives and eight reference hives), the MDD value would have been 6.4% (just before the tunnel phase).

To illustrate the sample sizes required to reach specific MDD values during the exposure phase, we also calculated power curves (Fig 7) using Monte Carlo randomization (for details, see S1 File). This has been done for the first day in the tunnel (28.4.), i.e. at a time when a pesticide would be applied.

Notably, while the variability was considerably reduced by the selection of hives after the pre-monitoring phase, there was a trend that variability gradually increased again towards the end of the study. This may be due to the variability in the use of foraging sites [32, 33] and (possibly as a consequence) foraging plants [34], which naturally introduce the variability between colonies over time. Since the use of MDD is a relatively new concept in honeybee risk assessment, there are only few studies available in which MDD were provided. In field study by Rolke et al. [35] eight equalized colonies with sister queens were used per study site and colony strength was estimated visually. MDD ranged between 15.2–21.4%. This is considerably higher than the MDD we observed after selection of colonies, despite of the smaller sample size used (four colonies instead of eight in Rolke et al [35]).

Using a larger number of hives and adding a monitoring phase to finally select a subset of colonies for the actual test does, however, increased the duration and workload for such a study. In particular the use of photography to count adult bees required a much larger number of field staff, since adult bee photography was done simultaneously for all hives to reduce bias by changing weather conditions. However, results have shown that colony size estimation by weight maybe a more cost effective alternative method with a similar level of measurement uncertainty.

Assessing forager activity and forager mortality with methods proposed in OECD (2007) [1] and using automated counting of foragers in videos a clear effect of the reference substance on the number of dead bees counted and on forager activity was observed. However, it is important to understand that the different methods for assessing mortality provide different information: The assessment of mortality according to OECD (2007) [1] and OEPP/EPPO

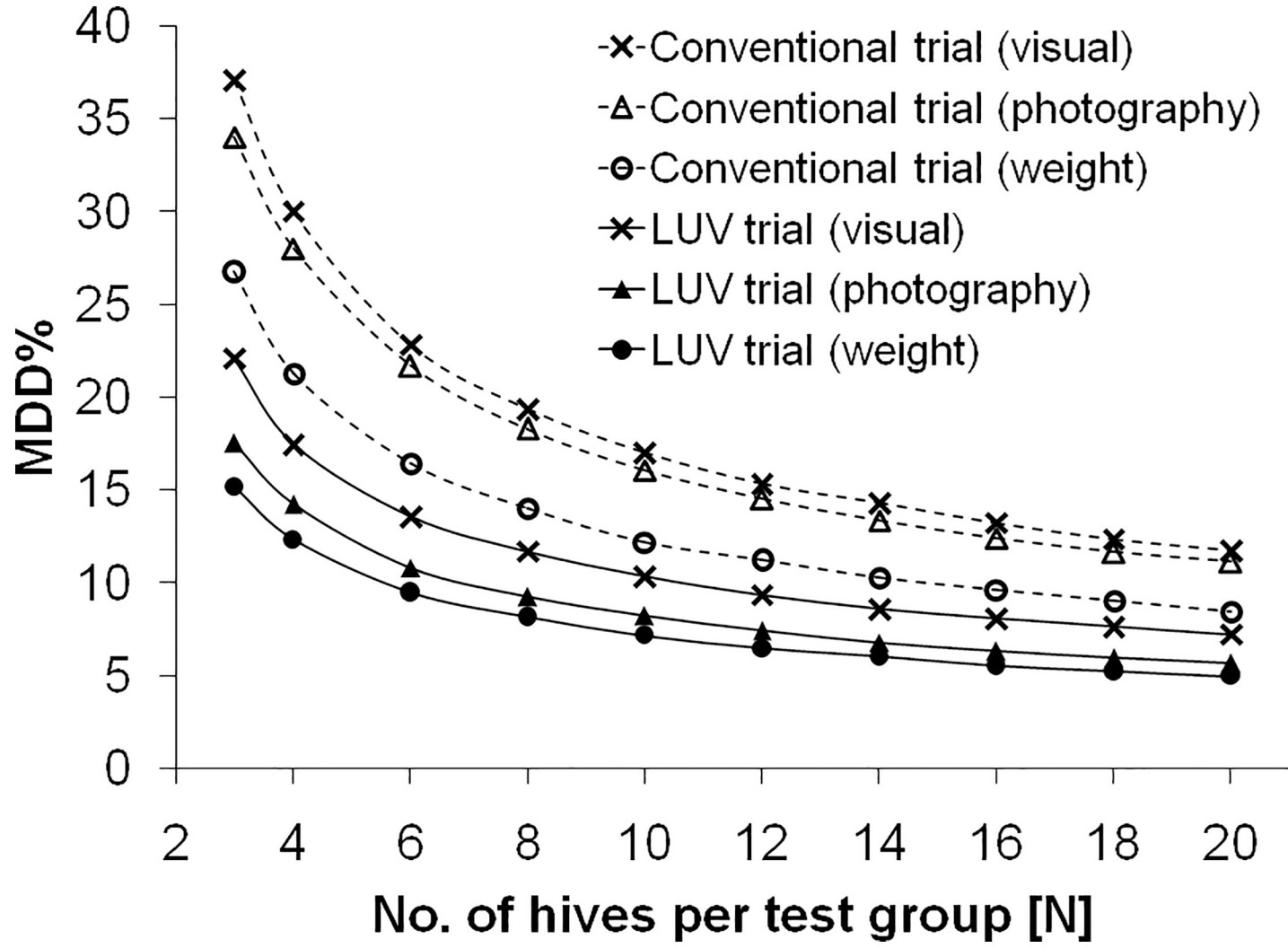

**Fig 7. Power curves for the first measurement day in the tunnel (28.4.) showing MDD% values and increasing number of hives per test group assuming either random selection of hives (conventional trial) or the selection of hives as conducted in the present LUV trial (for details see S1 File).**

(2010) [2] includes counting dead bees in dead bee traps in front of the hive and on plastic sheets placed in the tunnel. Dead bees found in dead bee traps are mostly bees that died within the hive and have been transported outside by conspecifics while dead bees collected on plastic sheets also include a considerable fraction foragers that died outside (in particular in a tunnel study). With regard to the Specific Protection Goals (SPGs) for forager mortality (e.g. a two-fold increased forager mortality over a period of three days is considered negligible; [3] it should be noted that dead bee traps do not provide a measure of forager mortality. Dead bees from plastic sheets probably partly reflect forager mortality. With video recordings, however, foraging activity can be estimated based on the number of bees entering and exiting the hives. Hence, this method more directly generates the data required to address the SPGs regarding forager mortality defined by EFSA [3]. Similarly, forager activity assessments as proposed in OECD (2007) [1] do not cover the whole forager activity of a colony but only a fraction limited by time and space. The new method of using automated video counts of bees entering and leaving the hives can be used to assess the whole forager activity of a colony for the whole foraging time from dawn to dusk.

To our knowledge this study is the first one assessing entire colonies, i.e. all cells of a hive were evaluated regarding the development of brood and of food stores throughout the study. This offered a unique insight into the development of hives and driving factors. Each hive included more than 3000 brood cells while in current brood trials only 100 to 200 cells containing eggs or larvae are assessed [1, 2]. To assess the impact of selection of a subset of brood cells on the results of a brood trial, either 100 or 300 cells with eggs were randomly chosen from one reference hive (hive 17–5) and BTR was calculated. This was repeated many times (see S1 File). BTR varied by about ±20% compared to the true BTR over all cells when choosing 100 cells with eggs for evaluation and by ±10% when choosing 300 cells with eggs. Hence, the selection of a small number of brood cells results in a considerable uncertainty of measured BTR.

Apart from removing uncertainty about the BTR in a given hive, the complete evaluation of all cells of the hives also made it possible to obtain insight into the factors that affect brood success and colony development. Low temperature of less than 12°C, which prevent foraging [36] coincided with a marked reduction of pollen stores and an increase in brood termination. Due to the importance of pollen as larvae food, the reduction of pollen stores results in an increase of open brood removal [37, 38]. 3D images of hives indicated also that brood termination depended on the location of pollen within the hive. In particular, pollen stores near brood cells determined brood success. This may be relevant for bee keeper practice and help to avoid colony losses. The maintenance of the frame location in the hive or an intentional relocation of frames with pollen stores could be used as a measure to increase brood success. The detailed information on brood success and the understanding of the factors increasing BTR may help to understand why BTR is sometimes very high in semi-field studies. For the future, this knowledge can help to decrease BTR in control hives, making this tests more reliable. In the past a number of meta-analyses have been conducted to understand why BTR in sometimes high in semi-studies [39, 40, 41] and less often in field studies [42] (but see also Candolfi et al. [43]). In these analyses more than 80 semi-field trials were analysed considering the factors season, weather, colony strength, tunnel size, larval and pupal mortality. Overall, there was a high variation of BTR and it was not very clear which factors correlate with BTR. While Pistorius et al. [39] found some influence of season (lower BTR in spring than in summer) and crop area (tunnel size) the results of the following analysis (which were partly based on the same trials) did not identify factors that very clearly affect BTR. In the present study, very detailed data on BTR were available: All brood cells of the hives were assessed (more than 3000 cells per hive), reducing variability of BTR which is due to sampling uncertainty and brood photography was conducted continuously over a period of more than two months (one month pre-monitoring, tunnel phase and one month post-tunnel phase). As a result $BTR_{egg}$ could be calculated for nine time points and for each frame side of each hive. The results indicate that $BTR_{egg}$ is determined by the ratio of pollen vs. open brood cells. Furthermore, also a clear spatial relation was found, i.e. mainly pollen stores on the same or opposite side of a frame were relevant for brood success, i.e. the pollen cells which can be easily accessed by nurse bees when feeding larvae. The influence of weather was clearly visible in the course of the study. Low temperature resulted in a rapid depletion of both pollen and nectar stores.

Concluding, with the presented LUV test, it was possible to considerably increase test-power (as reflected by MDD, which was reduced from a values >20% to 10.8% after selection of colonies). This may make it now possible to empirically determine Specific Protection Goals (e.g. the seven percent effect size regarding colony strength) recently proposed by EFSA [3, 7] using expert judgement, but also to test the toxicity of chemicals with a much higher certainty. Furthermore, new insights could be gained regarding the impact of weather and other biotic or abiotic factors can be studied in greater detail. However, as the results summarized above

are based only on one study, further testing may be required to verify the findings of this study.

## Supporting information

**S1 File.**
(PDF)

## Author Contributions

**Conceptualization:** Magnus Wang.

**Investigation:** Magnus Wang, Thiemo Braasch, Christian Dietrich.

**Methodology:** Magnus Wang.

**Supervision:** Magnus Wang.

**Validation:** Magnus Wang.

**Visualization:** Magnus Wang.

**Writing – original draft:** Magnus Wang, Thiemo Braasch, Christian Dietrich.

**Writing – review & editing:** Magnus Wang, Thiemo Braasch, Christian Dietrich.

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
