## [Decision Letter · Decision Letter 0]

10 Oct 2019

PONE-D-19-24362

Reduction of variability for the assessment of side effects of toxicants on honeybees and understanding drivers for colony development

PLOS ONE

Dear Dr Wang,

Thank you for submitting your manuscript to PLOS ONE. After careful consideration, we feel that it has merit but does not fully meet PLOS ONE’s publication criteria as it currently stands. Therefore, we invite you to submit a revised version of the manuscript that addresses the points raised during the review process.

Please address all the comments raised by both reviewers and, in particular, the comments on discussing causality and statistical power.

We would appreciate receiving your revised manuscript by Nov 24 2019 11:59PM. To enhance the reproducibility of your results, we recommend that if applicable you deposit your laboratory protocols in protocols.io, where a protocol can be assigned its own identifier (DOI) such that it can be cited independently in the future. For instructions see: http://journals.plos.org/plosone/s/submission-guidelines#loc-laboratory-protocols

We look forward to receiving your revised manuscript.

Kind regards,

James C. Nieh, Ph.D.

Academic Editor

PLOS ONE

**Journal Requirements:**

2. We note that one or more of the authors are employed by a commercial company: WSC Scientific GmbH.

**Comments to the Author**

1. Is the manuscript technically sound, and do the data support the conclusions?

Reviewer #1: No

Reviewer #2: Yes

2. Has the statistical analysis been performed appropriately and rigorously? 

Reviewer #1: No

Reviewer #2: Yes

3. Have the authors made all data underlying the findings in their manuscript fully available?

Reviewer #1: Yes

Reviewer #2: Yes

4. Is the manuscript presented in an intelligible fashion and written in standard English?

Reviewer #1: Yes

Reviewer #2: Yes

5. Review Comments to the Author

Reviewer #1: This study reports the results of a semi-field trial where caged colonies of honeybees (n = 4) are exposed to either exposure to an insecticide or control conditions. The authors use the data from the experiment to evaluate the power of the experimental design to detect treatment effects. The authors claim that the novel protocols used in their experiment enable fairly small treatment effects to be detected, which might be able to meet the levels of resolution stipulated in recent European regulations.

The experiments and data analysis appear to be technically sound and the paper is fairly well-written.

The main implication – that adjusted protocols will enable semi-field trials to detect pesticidal effects at the levels required by EU regulations – will be very important and interesting to a wide audience, including regulators, industry and environmentalists.

There are, however, some fairly major shortcomings that should be addressed if the wok is to reach its full impact, which I describe as follows.

1. Setting the baseline

It would be very useful to set the context – what is the MDD of previous semi-field trials? In the discussion, some evaluation of the level of improvement should be given.

2. Power analysis

One of the conventional components of a power analysis is an effort-power curve, which is the relationship between the MDD and sample size across a continuous range. Here it would be useful to see the curves from n = 2 to n = 20. In discussion, it would be useful to compare the power curves of the old protocols with the authors’ new protocols. Curves should be presented for all response variables.

3. Interpretation of causality

It is not correct to attribute the reduction in MDD over time to the selection of the hives because it might have decreased in any case. Justifying this assertion would require comparing the MDD when hives were picked from a pool of all hives versus the MDD when hives were picked from a reduced pool. Either do this or do not make the assertion.

4. Discussion

The current discussion is off-target. The main headline should be the improvement in MDD relative to past practice, which variables are most reliable, etc. The large opening section evaluating the various Delaplane methods does not warrant the space.

5. Data provision

There should be a table of means and SDs in the MS itself so that others can verify the power analyses.

Minor edits by line number

81-84: link to general theory by using some standard terms such as sampling, response variable.

90: decreased, surely?

100: clarify locations of phases 1-3

109: dissected = eliminated

386: ‘very small’ is subjective – currently the reference point is ‘almost negligible’ at 7%

Page 29: Figs 1 and 2 are indentical in my copy

Reviewer #2: The manuscript by Wang et al proposed and tested a new test design for risk assessment (RA) of pesticides in the field and semi-field, addressing major concerns.

The manuscript seems scientifically sound, mostly interesting for a relatively narrow community as the topic is very specific but with important broader outcome and global relevance as it proposes changes to the internationally used RA system. The MS could be more concise and clear, in terms of methodology and results/discussion. I think the MS is valuable and worth publishing in PlosONE after referee comments will be addressed. I highlight some major concerns that I believe, when addressed, could make the manuscript clearer, more impactful and robust.

*please address the CONS (negative aspects) related to your proposal, not only the pros. This is essential to estimate its feasibility and show a honest approach. i.e. increased costs in terms of expenses, time, organization, staff, etc.

*the proposed and current protocol should be reported in a table for clarity and ease of comparison. this tab can for ex. list the endpoints measured and how they are measured (time, etc) in your and the current system. (eg line 45-46, 33, 52,

*improve clarity of the methods/proposal: use multiple subchapters for each step i.e. in line 107 and after.

*"approximately" is often used (L214, 162, and many more). A more clear and exact description of your protocol is needed as this is required by guidances and gives solidity to your proposal. Add frequency of assessment times for all steps (ie L 174-176)

*SPG: how about the subelthal effects? Nonetheless you test forager activity etc, all other sublethal effects pesticides can cause are not addressed very much in your paper. This is another major concerns related to bee health, RA, SPGs. I'd clarify this and eventually state that your work goal does not include this specifically.

*clarify if you did field, semif, or both (ie. L90). certain basic details of your work should be easier to figure out. For ex did you test both scenarios with pre selection of colonies and not (L 212+)? If not you cannot compare the change in vairability .

*colony exclusion procedure (L 219) needs to be described and showed explicity in terms of methods and results. an explicit method for excludsion decision needs to be used and described (ie. decision threshold for each endpoint ie. varroa, etc?). This is a crucial point that RA needs to clarify explicitly and objectively. please refer to previous guidelines if available on the topic.

*the authors should address, briefly and concisely and explicitly in the discussion, how it was demonstrated, providing the key compartive quantitative values, that the LUV test improved the current standards (i.e. increase power); see lines 462-463. Please also explicitly report how you re results demonstrate that your assessment is more robust.

*major problems of field studies are not addressed: ie. the absence of real control colonies (i.e. pesticide-free) in the field (i.e. Campbell et al 2016 and Henry et al 2015 showed that control colonies were contaminated by the target pesticide too, and Tosi et al 2018 showed that the majority of the colonies in the environment are exposed to individual and multiple pesticides, even banned ones). I would at least mention this aspect and other key concerns related to field studies for RA.

LINE BY LINE COMMENTS:

abstract: spell out MDD

line 80. this was addressed before too. move above?

88 specify field or semifield?

412-415 video counts: you reported it causes higher variability. Please address this in this section and evetual other CONS.

416-418. I imagine there would be others, I'd double check.

discussion: text is very long, i think it should be shorter and more concise.

463-464: authors should be more careful when stating this, as you report results from 1 study, testing limited colonies and over a limited time frame (1 year). standard procedures for proposing new methods is ring testing them, ie. perfomed in multiple countries over multiple years. Thus, this statement seems not supported by your results.

figures: fig. 3: cannot see error bar in black bars.

references

Campbell, J. W., Cabrera, A. R., Stanley-Stahr, C. & Ellis, J. D. An evaluation of the honey bee (Hymenoptera: Apidae) safety profile of a new systemic insecticide, flupyradifurone, under field conditions in Florida. J. Econ. Entomol. 96, 875–878 (2016).

Henry, M. et al. Reconciling laboratory and field assessments of neonicotinoid toxicity to honeybees. Proc. R. Soc. B Biol. Sci. 282, 20152110 (2015).

Tosi, S., Costa, C., Vesco, U., Quaglia, G. & Guido, G. A 3-year survey of Italian honey bee-collected pollen reveals widespread contamination by agricultural pesticides. Sci. Total Environ. 615, 208–218 (2018).

6. PLOS authors have the option to publish the peer review history of their article (what does this mean?). If published, this will include your full peer review and any attached files.

Reviewer #1: No

Reviewer #2: No

---

## [Author Response · Author response to Decision Letter 0]

12 Nov 2019

Responses are included in file "Response to Reviewers.docx". Here is the same as text:

Journal Requirements:

Reply: We have taken these formatting requirements into account.

2. We note that one or more of the authors are employed by a commercial company: WSC Scientific GmbH.

Reply: We amended the financial disclosure statement as follows:

This research was funded entirely by the commercial company WSC Scientific GmbH. The funder provided support in the form of salaries for authors [MW, TB, CD] and materials, but did not have any additional role in the study design, data collection and analysis, decision to publish, or preparation of the manuscript. The specific roles of these authors are articulated in the ‘author contributions’ section.

Reply: We added the following Competing Interest Statement (we don’t expect a commercial benefit from this study, as industry is not interested in refined methodology (which may improve the detection of adverse effects of pesticides), however, academia or governmental authorities may be interested in an improvement of the method, which would, however, not result in any commercial benefit of the funding company):

[MW, TB, CD] are employed at WSC Scientific GmbH, which, amongst others, develops scientific software including software for the evaluation of honeybee trials. The purpose of this study was, however, not to promote such software, which contributes only marginally to the turnover of the company, but to develop better study design for the risk assessment. This does not alter our adherence to PLOS ONE policies on sharing data and materials. 

Reply: We have added these data in the Supplemental Material now and replaced “data not shown” with ”S1 Fig. 2, Tables 4-6”.

Comments to the Author

5. Review Comments to the Author

Reviewer #1: This study reports the results of a semi-field trial where caged colonies of honeybees (n = 4) are exposed to either exposure to an insecticide or control conditions. The authors use the data from the experiment to evaluate the power of the experimental design to detect treatment effects. The authors claim that the novel protocols used in their experiment enable fairly small treatment effects to be detected, which might be able to meet the levels of resolution stipulated in recent European regulations.

The experiments and data analysis appear to be technically sound and the paper is fairly well-written.

The main implication – that adjusted protocols will enable semi-field trials to detect pesticidal effects at the levels required by EU regulations – will be very important and interesting to a wide audience, including regulators, industry and environmentalists.

There are, however, some fairly major shortcomings that should be addressed if the wok is to reach its full impact, which I describe as follows.

1. Setting the baseline

It would be very useful to set the context – what is the MDD of previous semi-field trials? In the discussion, some evaluation of the level of improvement should be given.

Reply: We added a section in which we discuss a paper by Rolke et al. (2016), in which MDD was calculated (the use of MDD is a relatively new concept in pesticide risk assessment). Using twice as many colonies MDD ranged between 15.2–21.4% compared to about 10% in our study with only four colonies. 

2. Power analysis

One of the conventional components of a power analysis is an effort-power curve, which is the relationship between the MDD and sample size across a continuous range. Here it would be useful to see the curves from n = 2 to n = 20. In discussion, it would be useful to compare the power curves of the old protocols with the authors’ new protocols. Curves should be presented for all response variables.

Reply: Such a power curve would indeed be helpful. However, since honeybee trials according to OECD 75 are done with three to six colonies per test group at most, such data is not available (such studies are very cost intensive even with these sample sizes). We are not aware of any (published or unpublished) study with more than six colonies per test group.

3. Interpretation of causality

It is not correct to attribute the reduction in MDD over time to the selection of the hives because it might have decreased in any case. Justifying this assertion would require comparing the MDD when hives were picked from a pool of all hives versus the MDD when hives were picked from a reduced pool. Either do this or do not make the assertion.

Reply: We thank the reviewer for highlighting this. We added the information which MDD would have been reached without the selection procedure at the end of section “Pre-monitoring” in the results:

“We also compared MDD of colony strength by weight on the day of selection of colonies under the assumption of randomly sampling four colonies for each test group by Monte Carlo analysis. When randomly choosing hives MDD was on average 21.6%. I.e. the selection procedure achieved a reduction of MDD by a factor of about two (10.8% by the selection procedure vs. 21.6% by randomly choosing hives).”

4. Discussion

The current discussion is off-target. The main headline should be the improvement in MDD relative to past practice, which variables are most reliable, etc. The large opening section evaluating the various Delaplane methods does not warrant the space.

Reply: We shortened the section where the work by Delaplane was discussed considerably.

5. Data provision

There should be a table of means and SDs in the MS itself so that others can verify the power analyses.

Reply: We added this information to the Supplementary Information S1 (Tables 1-3).

Minor edits by line number

81-84: link to general theory by using some standard terms such as sampling, response variable.

Reply: We don’t understand this comment.

90: decreased, surely?

Reply: Thank you, yes it needs to be “decreased”.

100: clarify locations of phases 1-3

Reply: Thank you for highlighting this, it is indeed clearer if we mention the locations more clearly for phases 1 and 3 (for the tunnel phase this was already done). We added the location in parenthesis for phases 1 and 3 now.

109: dissected = eliminated

Reply: We replaced ‘dissected’ with ‘eliminated’

386: ‘very small’ is subjective – currently the reference point is ‘almost negligible’ at 7%

Reply: In the previous sentence we stated that effects of 10% were detectable with N=4, hence we feel it is fine to name these ‘very small’ (as the absolute value was provided). However, to make clear that we do not reach 7% yet with N=4 we changed the sentence to ‘rather small’. 

Page 29: Figs 1 and 2 are indentical in my copy

Reply: We corrected this now.

Reviewer #2: The manuscript by Wang et al proposed and tested a new test design for risk assessment (RA) of pesticides in the field and semi-field, addressing major concerns.

The manuscript seems scientifically sound, mostly interesting for a relatively narrow community as the topic is very specific but with important broader outcome and global relevance as it proposes changes to the internationally used RA system. The MS could be more concise and clear, in terms of methodology and results/discussion. I think the MS is valuable and worth publishing in PlosONE after referee comments will be addressed. I highlight some major concerns that I believe, when addressed, could make the manuscript clearer, more impactful and robust.

*please address the CONS (negative aspects) related to your proposal, not only the pros. This is essential to estimate its feasibility and show a honest approach. i.e. increased costs in terms of expenses, time, organization, staff, etc.

Reply: We added a paragraph highlighting the CONS of the applied methodology in the discussion:

“Using a larger number of hives and adding a monitoring phase to finally select a subset of colonies for the actual test does, however, increased the duration and workload for such a study by about a month. In particular the use of photography to count adult bees required a much larger number of field staff, since adult bee photography was done simultaneously for all hives to reduce bias by changing weather conditions. However, results have shown that colony size estimation by weight maybe a more cost effective alternative method with a similar level of measurement bias compared to photography.”

*the proposed and current protocol should be reported in a table for clarity and ease of comparison. this tab can for ex. list the endpoints measured and how they are measured (time, etc) in your and the current system. (eg line 45-46, 33, 52,

Reply: Such a table is already included in the supplemental information (Section “4. Comparison of study design in comparison to a conventional OECD 75 trial”), there the differences are compared for all endpoints measured. If the editor prefers we could move this table from the supplementary information to the manuscript? 

*improve clarity of the methods/proposal: use multiple subchapters for each step i.e. in line 107 and after.

Reply: We thank the reviewer for this excellent proposal. We have added a section before this line to first clarify which endpoints have been measured in which phase:

“Throughout the whole study the following endpoints were measured (see below for details on the methods): colony strength, content of all cells of all hives (incl. brood, nectar, pollen), dead bees and larvae in traps. Foraging activity by visual assessment was only conducted in the tunnel phase and forager activity and mortality measurements by videography were only conducted in the tunnel and post-tunnel phase.”

Furthermore we added a header “Step 1: Four week monitoring phase” and added a section for the following phase: “Step 2: A ten-day tunnel phase” and “Step 3: Post-tunnel phase”. We believe this does indeed help very much to clarify how the study was conducted.

*"approximately" is often used (L214, 162, and many more). A more clear and exact description of your protocol is needed as this is required by guidances and gives solidity to your proposal. Add frequency of assessment times for all steps (ie L 174-176)

Reply: We rephrased this section as follows (bold letters indicate changed text):

Line 162: “Measurements of flight activity were planned in weekly intervals during and after the tunnel phase using video recordings of the entrances of all hives over the entire activity phase (from dawn to dusk; see supplementary material). Actual dates varied by one to two days depending on weather conditions.”

Line 214: “Over a period of 32 days they were assessed approximately weekly (depending on weather conditions) to finally select eight very similar colonies […]”

*SPG: how about the subelthal effects? Nonetheless you test forager activity etc, all other sublethal effects pesticides can cause are not addressed very much in your paper. This is another major concerns related to bee health, RA, SPGs. I'd clarify this and eventually state that your work goal does not include this specifically.

Reply: We believe it is sufficiently clear that we do not address sublethal effects when discussing the specific protection goal (SPG) defined by EFSA, because whenever ‘SPG’ is mentioned the context is provided:

Line 404 in original manuscript: “With regard to the Specific Protection Goals (SPGs) for forager mortality […]”. Here it is clear that this only focuses on forager mortality.

Line 409: “Hence, this method more directly generates the data required to address the SPGs regarding forager mortality defined by EFSA (2013)” Same as above.

If the editor prefers we would of course be happy to add an additional sentence mentioning that sublethal effects are not considered. 

*clarify if you did field, semif, or both (ie. L90). certain basic details of your work should be easier to figure out. For ex did you test both scenarios with pre selection of colonies and not (L 212+)? If not you cannot compare the change in vairability.

Reply: Since the selection process would be the same for both field and semi-field trials, this sentence applies to both types of studies. We did, however, conduct only a semi-field study after the selection. But note that variability was reduced already before colonies were in the tunnels. In the sentence before it was already mentioned that we only did a semi-field study. However, to avoid any doubt we modified the sentence as follows:

“Results from our semi-field study demonstrate that variability of both field and semi-field trials can be decreased significantly with the proposed new test design and methodology.”

*colony exclusion procedure (L 219) needs to be described and showed explicity in terms of methods and results. an explicit method for excludsion decision needs to be used and described (ie. decision threshold for each endpoint ie. varroa, etc?). This is a crucial point that RA needs to clarify explicitly and objectively. please refer to previous guidelines if available on the topic.

Reply: We have added a more detailed description of the selection process in the methods section under the subheading “Selection of colonies at the end of the monitoring phase”. This section now reads as follows:

“From these eleven colonies eight colonies were selected and randomly assigned to the control and the reference group (four colonies per group). The selection had the aim of achieving colonies of similar strength during the tunnel phase. This same rationale is used in many laboratory toxicity trials, where very ‘similar’ animals, e.g. of similar age and strain are selected in order to decrease variability (and increase test-power). This was achieved based on the development of colony size throughout the pre-tunnel monitoring phase (as measured by weight), the number of capped brood cells per hive and similar mortality (measured in dead bee traps). Specifically, the selection was done in two steps: 1. Exclusion of colonies due to other reasons than colony size (e.g. high varroa count, untypically high mortality in dead bee traps). 2. Comparison of colony size during the monitoring phase, i.e. the most similar eight colonies in terms of colony strength were selected and assigned randomly to the two test groups.”

*the authors should address, briefly and concisely and explicitly in the discussion, how it was demonstrated, providing the key compartive quantitative values, that the LUV test improved the current standards (i.e. increase power); see lines 462-463. Please also explicitly report how you re results demonstrate that your assessment is more robust.

Reply: We modified the sentence:

“Concluding, with the presented LUV test, it was possible to considerably increase test-power.”

To:

“Concluding, with the presented LUV test, it was possible to considerably increase test-power (as reflected by MDD, which was reduced from a values >20% to 10.8% after selection of colonies).”

*major problems of field studies are not addressed: ie. the absence of real control colonies (i.e. pesticide-free) in the field (i.e. Campbell et al 2016 and Henry et al 2015 showed that control colonies were contaminated by the target pesticide too, and Tosi et al 2018 showed that the majority of the colonies in the environment are exposed to individual and multiple pesticides, even banned ones). I would at least mention this aspect and other key concerns related to field studies for RA.

Reply: While the issues mentioned by the reviewer are certainly an important point to consider in the risk assessment scheme in general, we feel that it is a bit out of the scope of this article. We could add a sentence such as: 

“As for all field or semi-field trials, there are still more issues to resolve for future testing, such as the use of truly unexposed control colonies (Henry et al., 2015; Campbell et al., 2016; Tosi et al., 2018). However, since colonies usually always have access to agricultural land or gardens when they are established before a field or semi-field test, it may be difficult to achieve this.” 

However, we feel it would be a bit lost in the discussion. If the editor prefers, we’d of course be happy to add at the end of the discussion.

LINE BY LINE COMMENTS:

abstract: spell out MDD

Reply: We did as suggested.

line 80. this was addressed before too. move above?

Reply: The methods above, described e.g. by Delaplane and Harbo (1987), differ to the ones we used. Delaplane and Harbo (1987) suggested to ‘create’ new colonies in order to achieve similar colonies, while we do not alter the colonies, but select the most similar ones. Therefore, we feel it is better to leave these parts of the text separately. 

88 specify field or semifield?

Reply: We added “semi-field” to be more precise.

412-415 video counts: you reported it causes higher variability. Please address this in this section and evetual other CONS.

Reply: This is probably a misunderstanding. We wrote that “the forager activity assessments as proposed in OECD (2007) [1] do not cover the whole forager activity of a colony but only a fraction limited by time and space.“ I.e. we consider the assessment using videos to be much more precise than the very short, visual assessments of only a few 1m² squares proposed by OECD.

416-418. I imagine there would be others, I'd double check.

Reply: We did not find any papers on studies in which entire colonies (all cells) were evaluated. There were only papers on methods to count capped cells (e.g. Colin et al., 2018).

Colin T, Bruce J, Meikle WG, Barron AB 2018. The development of honey bee colonies assessed using a new semi-automated brood counting method: CombCount. PLOS ONE: https://dx.plos.org/10.1371/journal.pone.0205816

discussion: text is very long, i think it should be shorter and more concise.

Reply: We tried to shorten the discussion, specifically paragraph 1. 

463-464: authors should be more careful when stating this, as you report results from 1 study, testing limited colonies and over a limited time frame (1 year). standard procedures for proposing new methods is ring testing them, ie. perfomed in multiple countries over multiple years. Thus, this statement seems not supported by your results.

Reply: We agree. We rephrased the paragraph and also added a sentence highlighting that so far this has been demonstrated only in one study (words in bold letters have been added):

“Concluding, with the presented LUV test, it was possible to considerably increase test-power (as reflected by MDD, which was reduced from a values >20% to 10.8% after selection of colonies). This does not only may make it now possible to empirically determine Specific Protection Goals (e.g. the seven percent effect size regarding colony strength) recently proposed by EFSA [3, 7] using expert judgement, but also to test the toxicity of chemicals with a much higher certainty. Furthermore, new insights cancould be gained regarding the impact of weather and other biotic or abiotic factors can be studied in greater detail. However, as the results summarized above are based only on one study, further testing may be required to verify the findings of this study.” 

figures: fig. 3: cannot see error bar in black bars.

Reply: We modified the figures with error bars so that also negative bars are now visible. 

References

Campbell, J. W., Cabrera, A. R., Stanley-Stahr, C. & Ellis, J. D. An evaluation of the honey bee (Hymenoptera: Apidae) safety profile of a new systemic insecticide, flupyradifurone, under field conditions in Florida. J. Econ. Entomol. 96, 875–878 (2016).

Henry, M. et al. Reconciling laboratory and field assessments of neonicotinoid toxicity to honeybees. Proc. R. Soc. B Biol. Sci. 282, 20152110 (2015).Tosi, S., Costa, C., Vesco, U., Quaglia, G. & Guido, G. A 3-year survey of Italian honey bee-collected pollen reveals widespread contamination by agricultural pesticides. Sci. Total Environ. 615, 208–218 (2018).

---

## [Decision Letter · Decision Letter 1]

6 Dec 2019

PONE-D-19-24362R1

Reduction of variability for the assessment of side effects of toxicants on honeybees and understanding drivers for colony development

PLOS ONE

Dear Dr Wang,

Thank you for submitting your manuscript to PLOS ONE. After careful consideration, we feel that it has merit but does not fully meet PLOS ONE’s publication criteria as it currently stands. Therefore, we invite you to submit a revised version of the manuscript that addresses the points raised during the review process.

Please provide in your revision the power curve requested by the reviewer. 

We would appreciate receiving your revised manuscript by Jan 20 2020 11:59PM. To enhance the reproducibility of your results, we recommend that if applicable you deposit your laboratory protocols in protocols.io, where a protocol can be assigned its own identifier (DOI) such that it can be cited independently in the future. For instructions see: http://journals.plos.org/plosone/s/submission-guidelines#loc-laboratory-protocols

We look forward to receiving your revised manuscript.

Kind regards,

James C. Nieh, Ph.D.

Academic Editor

PLOS ONE

Reviewers' comments:

Reviewer's Responses to Questions

**Comments to the Author**

1. If the authors have adequately addressed your comments raised in a previous round of review and you feel that this manuscript is now acceptable for publication, you may indicate that here to bypass the “Comments to the Author” section, enter your conflict of interest statement in the “Confidential to Editor” section, and submit your "Accept" recommendation.

Reviewer #1: (No Response)

2. Is the manuscript technically sound, and do the data support the conclusions?

Reviewer #1: Yes

3. Has the statistical analysis been performed appropriately and rigorously? 

Reviewer #1: Yes

4. Have the authors made all data underlying the findings in their manuscript fully available?

Reviewer #1: Yes

5. Is the manuscript presented in an intelligible fashion and written in standard English?

Reviewer #1: Yes

6. Review Comments to the Author

Reviewer #1: Your revisions are largely satisfactory and I think it will be a valuable contribution to this under-investigated and important topic. I have one request that was not addressed and I point out some minor edits.

1. Power curve - as previously requested

Since you are able to use Monte Carlo randomization, it should be straightforward to estimate power curves by sampling from parametric distributions with the same means and variances as the observed data. Or use the method that you used to estimate the effect of doubling the sample size (lines 410-413). Obviously, it would be nice to generate these empirically, but you can go a long way using your present observation.

The really valuable contribution here is to estimate the number of hives needed to detect 7% effects currently specified by the Specific Protection Goals and to indicate the likely number of hives needed to detect any given effect size.

Minor edits by line number:

32: at driving?

57: forager

68: certainly not 'impossible' - 'logistically demanding', perhaps

77: comma after colonies

143: 'toxicant application' - even 'reference substance' is too specialist jargon for this journal

209: based on

258: define the pool that the selection was made from (the 11 colonies with or without replacement - with, surely?)

266: t-tests of the effect of toxicant exposure on ..

275 - see 143

286 - as 143

7. PLOS authors have the option to publish the peer review history of their article (what does this mean?). If published, this will include your full peer review and any attached files.

Reviewer #1: No

---

## [Author Response · Author response to Decision Letter 1]

8 Jan 2020

Response to reviewers

Reviewer #1: Your revisions are largely satisfactory and I think it will be a valuable contribution to this under-investigated and important topic. I have one request that was not addressed and I point out some minor edits.

1. Power curve - as previously requested

Since you are able to use Monte Carlo randomization, it should be straightforward to estimate power curves by sampling from parametric distributions with the same means and variances as the observed data. Or use the method that you used to estimate the effect of doubling the sample size (lines 410-413). Obviously, it would be nice to generate these empirically, but you can go a long way using your present observation.

Reply: Thank you for this suggestion, we thought you had meant to do a power analysis with actual data. Using Monte Carlo instead is a good idea. We added power curves as suggested as Figure 7 to the main text in the location proposed by the reviewer. Details to the methods used are shown in the supplementary materials (section 6).

The really valuable contribution here is to estimate the number of hives needed to detect 7% effects currently specified by the Specific Protection Goals and to indicate the likely number of hives needed to detect any given effect size.

Reply: We agree, this is now shown in Figure 7.

Minor edits by line number:

32: at driving? 

Reply: Thank you, this is a typo it must be “and driving”

57: forager 

Reply: We changed this to “foragers”

68: certainly not 'impossible' - 'logistically demanding', perhaps

Reply: As we refer to evaluations by EFSA of data that were generated in the past, we think that it is correct to state that it is impossible to detect small effects using these data. The reviewer probably means that in principle one could detect small effects with more effort. We would agree, but this is not meant here. 

77: comma after colonies 

Reply: Thank you, we added a comma.

143: 'toxicant application' - even 'reference substance' is too specialist jargon for this journal 

Reply: We rephrased the sentence as follows: “Toxicant application (reference substance) was conducted two days after the start of the tunnel phase.”

209: based on 

Reply: Thank you, we added “on” 

258: define the pool that the selection was made from (the 11 colonies with or without replacement - with, surely?) 

Reply: We added “with replacement” in backets.

266: t-tests of the effect of toxicant exposure on .. 

Reply: We added “the effect of toxicant exposure on” as suggested. 

275 - see 143 

Reply: We have rephrased the sentence as follows (but we feel it is less clear now, the term ‘reference group’ is a term that all ecotoxicologists will be familiar with): “After the selection of the colonies, four hives were randomly assigned to each the control group and the group in which a toxicant was applied (reference group), respectively and they were placed into the tunnels.” 

286 - as 143

Reply: We have replaced “control” with “control group”.

---

## [Decision Letter · Decision Letter 2]

4 Feb 2020

Reduction of variability for the assessment of side effects of toxicants on honeybees and understanding drivers for colony development

PONE-D-19-24362R2

Dear Dr. Wang,

We are pleased to inform you that your manuscript has been judged scientifically suitable for publication and will be formally accepted for publication once it complies with all outstanding technical requirements.

With kind regards,

Nicolas Desneux

Academic Editor

PLOS ONE

Additional Editor Comments (optional):

Reviewers' comments:

Reviewer's Responses to Questions

**Comments to the Author**

1. If the authors have adequately addressed your comments raised in a previous round of review and you feel that this manuscript is now acceptable for publication, you may indicate that here to bypass the “Comments to the Author” section, enter your conflict of interest statement in the “Confidential to Editor” section, and submit your "Accept" recommendation.

Reviewer #1: All comments have been addressed

2. Is the manuscript technically sound, and do the data support the conclusions?

Reviewer #1: Yes

3. Has the statistical analysis been performed appropriately and rigorously? 

Reviewer #1: Yes

4. Have the authors made all data underlying the findings in their manuscript fully available?

Reviewer #1: (No Response)

5. Is the manuscript presented in an intelligible fashion and written in standard English?

Reviewer #1: Yes

6. Review Comments to the Author

Reviewer #1: This is nicely completed. All of my concerns/corrections have been addressed and the power curve for MDD will be useful for practitioners and regulators.

7. PLOS authors have the option to publish the peer review history of their article (what does this mean?). If published, this will include your full peer review and any attached files.

Reviewer #1: No

---

## [Editor Report · Acceptance letter]

5 Feb 2020

PONE-D-19-24362R2 

Reduction of variability for the assessment of side effects of toxicants on honeybees and understanding drivers for colony development 

Dear Dr. Wang:

I am pleased to inform you that your manuscript has been deemed suitable for publication in PLOS ONE. Congratulations! Your manuscript is now with our production department. 

With kind regards,

on behalf of

Dr. Nicolas Desneux 

Academic Editor

PLOS ONE